# The Epidemiological Characteristics of the COVID-19 Pandemic in Europe: Focus on Italy

**DOI:** 10.3390/ijerph18062942

**Published:** 2021-03-13

**Authors:** Giovanni Gabutti, Erica d’Anchera, Francesco De Motoli, Marta Savio, Armando Stefanati

**Affiliations:** 1Department of Medical Sciences, University of Ferrara, via Fossato di Mortara 64/B, 44121 Ferrara, Italy; sta@unife.it; 2Department of Medical Sciences, Postgraduate School of Hygiene and Preventive Medicine, University of Ferrara, 44121 Ferrara, Italy; dncrce@unife.it (E.d.); dmtfnc@unife.it (F.D.M.); svamrt@unife.it (M.S.)

**Keywords:** COVID-19, SARS-CoV-2, epidemiology, preventive measures, public health, nursing homes, intensive care unit, mortality, lethality, R0

## Abstract

Starting from December 2019, SARS-CoV-2 has forcefully entered our lives and profoundly changed all the habits of the world population. The COVID-19 pandemic has violently impacted the European continent, first involving only some European countries, Italy in particular, and then spreading to all member states, albeit in different ways and times. The ways SARS-CoV-2 spreads are still partly unknown; to quantify and adequately respond to the pandemic, various parameters and reporting systems have been introduced at national and European levels to promptly recognize the most alarming epidemiological situations and therefore limit the impact of the virus on the health of the population. The relevant key points to implement adequate measures to face the epidemic include identifying the population groups most involved in terms of morbidity and mortality, identifying the events mostly related to the spreading of the virus and recognizing the various viral mutations. The main objective of this work is to summarize the epidemiological situation of the COVID-19 pandemic in Europe and Italy almost a year after the first reported case in our continent. The secondary objectives include the definition of the epidemiological parameters used to monitor the epidemic, the explanation of superspreading events and the description of how the epidemic has impacted on health and social structures, with a particular focus on Italy.

## 1. Introduction

The COVID-19 epidemic, after the initial spread starting from the city of Wuhan in the province of Hubei in China, has begun to spread to other continents, first Europe, violently hitting member states and putting a strain on all health systems. [1]. Even before the pandemic was declared by the World Health Organization (WHO), the cases of COVID-19 were already beginning to significantly increase in some European states and then ended up involving the continent as a whole. As of 12 January 2021, 84,532,824 confirmed cases were recorded in the world; in the European Union and the European Economic Area (EU/EEA) 15,857,298 cases and 1,845,597 deaths were recorded [2].

The main objective of this work is to summarize the epidemiological situation of COVID-19 pandemic in Europe and Italy almost a year after the first reported case in our continent. The secondary objectives include the definition of the epidemiological parameters used to monitor the epidemic, the explanation of superspreading events and the description of how the epidemic has impacted on health and social structures, with a particular focus on Italy.

## 2. Epidemiological Parameters and Superspreading Events

Several parameters can be used to appropriately describe the spreading dynamics of an infectious disease; among these, the fundamental one is the basic reproduction number (R0), which represents the average number of secondary infections produced from each infected individual in a completely susceptible population. A value of R0 >1 assumes that the pathogen is spreading and a value <1 indicates an epidemic potentially capable of being contained. R0 is a value to be considered as a function of the number of contacts of an infected person and the duration of the infectivity. This indicator alone is not enough to adequately describe the spreading of the disease. Another factor, called Rt, helps make this description; its definition is equivalent to that of R0 with the difference that Rt is calculated over time. Rt allows, for example, to monitor the effectiveness of interventions adopted during an epidemic. R0 and Rt can be calculated from daily case incidence. R0 and Rt differ from each other; they are complementary and not contradictory. In Italy, R0 and Rt are used to monitor the epidemiological trend of COVID-19. The Italian National Health Agency (Istituto Superiore di Sanità, ISS) underlines that the two indicators are calculated on slightly different data: Rt is calculated on the subgroup of non-imported cases and refers to when these symptoms developed (considered by date of early symptoms). The ISS decided to consider for the calculation of both R0 and Rt only symptomatic cases. The statistical method of calculating Rt is defined as robust if it is calculated on a number of infections identified according to sufficiently stable criteria in time. Criteria adopted in all Italian regions to identify symptomatic cases or hospitalize most serious cases are almost the same, and the number of this type of patients can be closely linked to the transmissibility of the virus. On the other hand, the identification of asymptomatic infections depends a lot on the ability of public health departments to perform screening tests and, unfortunately, this can vary a lot over time, especially in the case of overloading of the health system [3,4].

Another particularly effective indicator in the epidemiological description of the COVID-19 pandemic is the secondary attack rate (SAR), which defines the ability to become infected in groups of susceptible individuals linked by close contacts, such as families or in overcrowded spaces. Some studies during the COVID-19 pandemic have assumed that the secondary attack rate can be as high as 35% if secondary cases originating from a known index case in a restricted setting, such as a restaurant dinner or a family lunch, are considered. In such situations, it is therefore of primary importance to promptly trace the close contacts of the participants in these events to effectively reduce the transmission of the virus [4,5].

These indicators can be supported by a further index, called factor K (coefficient of dispersion K), useful for quantifying the individual differences of a single infected subject to be able to transmit the infection to a susceptible contact. In practice, some subjects are able to infect a large number of people while others have a lower ability to spread the virus; a factor of K < 1 is indicated if the variation between the different subjects is large, while K > 1 if it is not [4]. The factor K is of crucial importance for the correct framing of the so-called super spreader events (SSEs) relatively common in infectious diseases’ epidemiology. Outbreaks sustained by an infectious subject infecting several contacts has been described not only for COVID-19 but also for other infections such as tuberculosis and measles. SSEs happen and are documented in coffee shops, restaurants, churches and ships, as well as in health and social settings. In these cases, R0 is not useful to justify the total number of infections [4,6].

Regarding the COVID-19 pandemic, several SSEs have been reported, highlighting how many factors have a role in the spread of SARS-CoV-2. Taking into account the direct, person-to-person, diffusion of the virus, there are individual factors that contribute to the spread of viral particles such as speech characterized by a greater production of saliva and therefore of droplets, the viral load, the site of greatest concentration of the virus (for example, upper airways) and the availability of an infecting viral dose. Other ways of viral spreading imply environmental factors such as staying in closed unventilated spaces, the persistence of the virus on surfaces, the population density, the non-compliance rather than the compliance with preventive regulations such as social distancing, hand hygiene and the use of personal protective equipment. Finally, some social parameters must be considered, such as the type of work (health workers in close contact with several people) and the lifestyle of the subject, who may be more or less prone to close contacts with different people. For preventive purposes, it is absolutely relevant the early identification of cases during these events and to quickly carry out a contact tracing, including a retrospective investigation also of all contacts occurred before the index event in order to effectively slow down the spread of SARS-CoV-2 [5,6,7].

## 3. European and Italian Surveillance Systems

The European Center for Disease Prevention and Control (ECDC) has established community surveillance objectives for COVID-19 that can reconcile both the extreme usefulness of having updated data, useful to modify the strategies for containing the pandemic, and not to overload already overwhelmed health systems [8] (Table 1).

The ECDC recognizes the real difficulty that European countries have in daily updating data and asks for at least a weekly report with aggregated data on the total number of cases, a reduced dataset (case-based) on the most severe cases in at risk groups and a weekly update on viral sequences. It also indicates, in order of relevance, the list of groups to be tested as a priority in the case of limited resources, identifying health workers and the elderly as priority groups, followed by hospitalized patients with respiratory symptoms and finally all cases with even mild symptoms [8].

In Italy, national surveillance was arranged starting from 22 January 2020 and the first criteria and reporting methods to the ISS Department of Infectious Diseases were defined, identifying supervisors at regional level and in each autonomous province. The ISS designed a dedicated computer platform where data had to be sent, collected and analyzed at national level. Unfortunately, notification to ISS is available only 2–3 weeks after the onset of the disease. In almost all cases, the swab for diagnosis is carried out after the onset of first symptoms. This excludes the evaluation of the incubation period and only in some cases a positive swab for SARS-CoV-2 is obtained from close contacts of a patient. In addition, the speed of swab execution largely depends on local organization, significantly different throughout the Italian territory due to demand and the availability of medical staff as well as of laboratories [9].

## 4. Epidemiological Characteristics of SARS-CoV-2 Infection in Europe

Europe was the first continent heavily hit by SARS-CoV-2 pandemic after Asia. As of 6 December 2020, total cases in the European continent were 20,296,503 and deaths 449,793. Of these, 56% were men and 89% were >65 years of age. In addition, 96% of people who died from COVID-19 suffered from at least one other pathological condition. In particular, 82% of deceased subjects had a cardiovascular disease. The eight countries with the highest number of cases are, in decreasing order: Russia, France, Italy, the United Kingdom, Spain, Germany and Poland, with 12%, 11%, 8%, 8%, 8%, 6% and 5% of cases, respectively. As regards the total number of deaths, the United Kingdom ranks first (14% of total deaths), followed by Italy, France, Spain and Russia with 13%, 12%, 10% and 10%, respectively [10].

In the EU/EEA region, the first cases of COVID-19 were identified at the end of January 2020. In France and Finland, cases were imported while in Germany a local transmission occurred with an indirect epidemiological connection with Wuhan [11]. Following these first notifications, Italy began to record a significant increase in COVID-19 cases due to the clusters identified in the Veneto and Lombardy regions at the end of February [12]. Despite an initial and heavy involvement of Italy alone (February and March), SARS-CoV-2 considerably spread in other countries of the old continent. During the first week of March, 62% of cases in Europe were notified by Italy; within one month, in the first week of April, Spain reported 21%, Italy 20%, Germany 15% and France 11% of cases in Europe and then the whole continent was involved [13,14]. Since mid-March 2020, the number of new diagnosed cases in the EU/EEA region considerably started to rise, accounting for an increasing rate of non-Chinese cases; European countries begun to move rapidly towards a sustained community transmission framework [15]. With the widespread presence of SARS-CoV-2 at the community level, it was necessary to implement several preventive measures, both collective (restrictive “stay at home” policies recommended or mandatory) and individual (hand hygiene, respiratory hygiene and social distancing). Restrictive measures adopted have not been univocal and concurrent in all European countries; in any case, they led to a stabilization of the incidence. In fact, at the end of April 2020, the initial transmission wave had exceeded its peak in twenty EU/EEA countries and a decreasing incidence was reported (Figure 1) [16].

After a partial relief in summer, a resumption of incidence was observed during the first week of July; however, there was not a unique explanation for this resurgence of cases in different countries [17]. From the end of August, a further and increasingly substantial rise in COVID-19 cases was observed throughout the EU/EEA region, initially in young people and later also in adults and elderly. The continuous growth of positivity index highlighted how this increase was no longer related only to the higher number of tests performed or to the modified case definition but to a real increased transmission of SARS-CoV-2 (Figure 1) [18]. Noteworthy, during spring, the most affected age group was that of the >65-year-olds while from July 2020 the notification rates significantly increased in the younger age classes (15–24 years and 25–49 years) [19]. 

The excess mortality related to COVID-19 during the first wave was particularly relevant in Belgium, Italy, France, Ireland, Spain, the Netherlands, Sweden, Switzerland and the United Kingdom. During the second wave, an excess of mortality was seen in Slovenia and confirmed for the above-mentioned countries except for Sweden, Ireland and the United Kingdom; the highest excesses mortality was found in ≥75-year-old subjects [19].

Despite the strong impact that COVID-19 had on population, in terms of both cases and deaths (Figure 1), the level of herd immunity in most European nations at the end of October was <15%, highlighting how the risk for the population was already high before the winter season [18]. Since the end of October many European countries have begun to implement and gradually re-establish the restrictive measures, previously already adopted during the first months of the pandemic. Despite this, there has been a constant and relevant increase in the transmission of SARS-CoV-2 as well as in COVID-19 cases, particularly noteworthy in respect to levels reached during the first part of summer [20].

The difficulty of facing again a situation comparable to the first months of the pandemic has not only put a strain on health systems of all countries but also on the willingness and ability to face a worsening emergency of the whole European population. This phenomenon, characterized by the difficulty of accepting, again and positively, the restrictive containment measures, is identified as “pandemic fatigue” [20]. Besides, the recent discovery of several variants of SARS-CoV-2 has raised new concerns in the scientific community. The presence of viral variants was expected, and there are currently several thousand of them, most of which have not created any further threat to humans. Unfortunately, there are also some concerning variants; the most worrying are the Danish (in mink), English, South African and Brazilian variants [21,22]. The role of an intermediate animal in the spread of SARS-CoV-2 is widely discussed and the infection has been defined as a zoonosis [23]. Although the initial stages of the pandemic were traced back to a Wuhan wet market and several tests carried out on the cages and on the surfaces of the market benches were SARS-CoV-2 positive, no tests carried out on animals were positive [24]. Nevertheless, SARS-CoV-2 transmission between different animal species is considered a possible event. An example is the epidemic detected in mink starting from April 2020 in Denmark. In this case, transmission can be bi-directional, from human to mink and from mink to human [25]. It is currently believed that mink-related viral variants do not have greater transmissibility or a more severe clinical course [25].The only fearsome variant is the so-called “Cluster 5”, as it seems to have a lower antigenicity, and thus a lower chance to be identified by diagnostic tests. It has been postulated that the presence of these new variants could be related to transmission of SARS-CoV-2 by humans to mink, followed by viral mutations and subsequently transmission back to humans [25]. In general, the SARS-CoV-2 ability to infect animals depends on several factors, unfortunately not yet fully understood. Among these, the compatibility between the viral protein S and the host ACE-2 receptor is included [25]. Concerning the English variant (Variant of Concern, VOC 202012/01), the hypothesis of human–animal–human spreading seems very unlikely. The hypothesis that SARS-CoV-2 was able to progressively accumulate mutations is not widely accepted; in fact, the variant has an unusually high number of mutations of the spike protein and has new genomic properties, compared to the scheme of random mutations forecasted by models. A possible hypothesis concerns the genesis of the variant from a single individual with “long-lasting” infection and compromised immune system. This fact would have allowed SARS-CoV-2 to accumulate favorable mutations with the aim of evading the immune system [22]. The greater transmissibility of this variant is of concern; it has been estimated that R0 for the English variant has increased by 0.4% or more with an increase in transmissibility equal to 70% [26]. The first cases caused by this variant were recorded in September 2020, but a significant and sudden increase in cases in the southern region of England was registered in November. Most cases involve <60-year-old subjects [26] and at the end of December an increase in the severity of the clinical course has not yet been found [22]. Unfortunately, cases due to this variant are not limited to the United Kingdom but have also been found in other European (Belgium, Denmark, Finland, France, Germany, Iceland, Ireland, Italy, The Netherlands, Norway, Portugal, Spain and Sweden) and not European countries (Australia, Canada, Hong Kong, India, Israel, Japan, Jordan, Lebanon, South Korea, Switzerland and Singapore) [22]. In addition, a third variant of SARS-CoV-2, the South African variant (501.V2), has been circulating since last August, and it is worrisome due to its greater transmissibility [22]. Finally, a fourth variant needs to be monitored: the Brazilian (P.1). To date, it has only been identified in Brazil and in travelers from Brazil. The risk for a spread in Europe of the variants mentioned above is currently considered very high [21,22]. Besides, in almost all EU/EEA countries, except Denmark, the ability to track and identify the spread of SARS-CoV-2 variants is very limited [22]. As a matter of fact, since September 2020, only Denmark and Norway have sequenced and published more than 1% of cases and only eight countries have sequenced and published 0.1% of cases related to these new variants [22]. The SARS-CoV-2 positivity of some gorillas has also recently been reported in the San Diego Zoo in the United States. It has been speculated that animals became infected due to the presence of some positive zoo operators; if so, this would be the first confirmed SARS-CoV-2 transmission between primates [27].

The start of vaccination campaigns across Europe at the end of December certainly brought a glimmer of hope; the epidemiological trend should positively reflect the rate of vaccinated people which increases every day. As of 14 February 2021, 21,944,944 doses have been administered in the EU/EEA countries [28]. However, several doubts are arising on the management of vaccinated subjects in case of contact with a positive case. The American Center for Disease Control recommends in a recently published document not to quarantine these subjects as long as they have completed the vaccination course (with the m-RNA vaccine) within no more than three months and have no symptoms [29]. There is currently no such recommendation in Europe, but it is certainly an issue that must soon be addressed and clarified. Unfortunately, the lack of data on the ability of vaccinees to transmit the virus makes it difficult to provide complete indications.

**Figure 1 ijerph-18-02942-f001:**
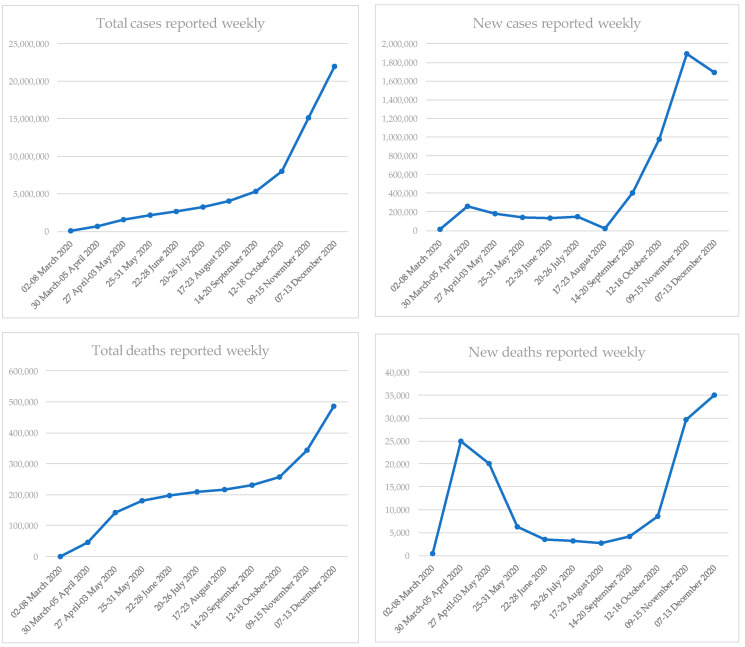
Trend of COVID-19 cases and deaths in EU/EEA region, 2 March 2020–13 December 2020 (modified from [13,14,30,31,32,33,34,35,36,37,38]).

## 5. Epidemiological Characteristics of SARS-CoV-2 Infection and COVID-19 in Italy

### 5.1. General Epidemiology in Italy

In Italy, according to the latest update of the ISS of 22 December 2020, the SARS-CoV-2 epidemic recorded a total number of cases equal to 1,963,023 (Figure 2) [39,40] out of a total population of 60,244,269 as reported by the ISTAT statistical yearbook published in 2020 [41]. The impact of the second wave in terms of total diagnosed cases and, albeit to a lesser extent, also of total symptomatic cases, is clearly higher than that of the first wave, mainly due to the greater number of swabs performed, although from mid-November a decline has been observed. The median age of those affected by the virus is equal to 48 years (range 0–109 years), a downward trend compared to the one recorded during the first two months of the epidemic (60 years). The lowest value was registered in mid-August with a median age of about 30 years. From the month of May, there was a decrease in cases in all age groups, more markedly in subjects >50 years of age, which however showed a new increase from mid-August. Since the end of September, cases considerably increased, especially in the age groups 0–18 and 19–50 years, while from mid-November there was again a decrease in all age groups. Females are the mostly affected gender (52.5%), while males were mostly involved in the initial phase of the epidemic. Looking at the total number of reported cases, the date of appearance of the first symptoms is available for 1,037,226 out of 1,963,023 subjects. This gap is linked both to the fact that many subjects were asymptomatic and to the consolidation of data after some time. The time elapsed between diagnosis and the date of onset of symptoms is available for 1,036,238 cases; from the beginning of the epidemic, this ranged 4–5 days until mid-June, and then it decreased to two days, although the decline was not regular over the months [39]. In Italy, notifications during the first wave reached a peak in March with 113,011 diagnoses; then, a progressive decline in April (94,257) and May was registered, mainly as a result of the national lockdown announced on 11 March 2020 [42].

### 5.2. Cases Stratified by Regions

During the early stages of epidemic, the geographic spread was very heterogeneous; for this reason, the Italian provinces were divided into three groups accordingly to the distribution of standardized incidence rates (“low”, “medium” and “high” spread). The low prevalence provinces (34, mainly those of the South and the Islands) are those with a rate <60 cases per 100,000 residents; the medium-spread provinces (32, mostly those of Central Italy) have a rate between 60 and 150 cases per 100,000 residents; and the high circulation group includes those provinces (41, especially in Northern Italy) with a rate >150 cases per 100,000 residents. [42]

As of 22 December 2020, the regions that reported the highest number of cases were Lombardy, Veneto and Piedmont with 469,991 (23.9%), 222,868 (11.4%) and 186,045 cases (9.5%), respectively. On the contrary, the regions with the lowest number of subjects affected by the virus were Molise, Aosta Valley and Basilicata with 6139 (0.3%), 7096 (0.4%) and 9864 cases (0.5%), respectively. It should be noted that the Aosta Valley, while reporting a not particularly high number of cases, has a cumulative incidence per 100,000 inhabitants even higher than that of Lombardy (5654.14 vs. 4651.55 per 100,000). In addition, the autonomous province of Bolzano, with a cumulative incidence of 5183.99 per 100,000 inhabitants and a total of 27,583 diagnosed cases, precedes Lombardy (Table 2). Most (87.1%) cases were diagnosed in 10 regions: Lombardy, Veneto, Piedmont, Campania, Emilia-Romagna, Lazio, Tuscany, Sicily, Puglia and Liguria [39].

There are several possible causes at the basis of a faster and greater spread of the virus in the regions of Northern Italy (more central geographical location, greater migratory flows from abroad, less time, resources and the absence of protocols to better face the emergency in the early stages). A detailed discussion of this is beyond the scope of this manuscript; however, all of this cannot be reduced to a simple difference in terms of population density, as shown in Table 2 [41].

### 5.3. Severity of the Clinical Course

One of the main differences observed between the first weeks after the onset of the epidemic and the following months concerns the course and the severity of the infection. In fact, while initially there was a higher rate of severe cases and deaths already at the time of diagnosis (swabs carried out post-mortem), subsequently an increase in asymptomatic or pauci-symptomatic cases was observed as well as a marked reduction of severe cases and deaths [39].

Figure 3 shows the trend of hospitalizations due to COVID-19 in the period 20 February–22 December 2020. The distribution is biphasic with the first peak in the early stages of epidemic (March), while the second is in November. It is worth noting that data for the month of December may still be incomplete due to an update delay [40].

### 5.4. Mortality and Lethality

Deaths updated to 22 December 2020 account for 67,540 [39]. The distribution of COVID-19 mortality presented a double peak over the year: the first in the period March–May and the second between October and December. A sharp decline was observed during the summer season (Figure 4) [40]. Based on the ISS report on deceased patients updated to 16 December 2020, the region with the highest number of deaths was Lombardy with 24,070 cases (37.9%), followed by Emilia-Romagna (6718, 10.6%) and Piedmont (5543, 8.7%). Calabria, Basilicata and Molise were the regions with the lowest number of deaths, 325 (0.5%), 174 (0.3%) and 170 (0.3%), respectively. The average age of deceased subjects was 80 years (median 82); 42.4% of deaths involved women (25,185). The median age was equal to 85 and 80 years in women and men, respectively. Data concerning the age differed considerably from that concerning the above-mentioned median age of diagnosis; the total number of deceased subjects <50 and <40 years of age was 737 and 190, respectively. Data on pre-existing pathologies were available from 5962 deceased patients after hospitalization; the mean number of diseases observed in this population was 3.6. Hypertension was the most frequent condition found in both genders (68.7% in females vs. 64.2% in males). Women with dementia and type II diabetes mellitus rank second and third place (32.1% and 26.7%), followed by atrial fibrillation (25.6%) and ischemic heart disease (23.4%). In men, type II diabetes mellitus and ischemic heart disease showed overlapping rates (30.7% and 30.8%), followed by atrial fibrillation (23.4%) and chronic renal failure (22.2%) [43].

The lethality of the disease showed a relevant increase according to age, reaching the maximum in the ≥90 years age class (36.0% in males and 20.4% in females). Total lethality was 4.1% in men and 2.9% in females. Lethality was also higher in males of all age groups; the lethality value was close to 0 in both genders in the 0–29 years age group. As of 22 December 2020, the healed subjects were 1,151,639 [39].

Taking into account the total number of deaths for all causes, the provisional toll for the period January–November 2020 amounted to 664,623 deaths, i.e., 77,136 more deaths than the average registered in the period 2015–2019. On the other hand, considering only the period when deaths from COVID-19 are included (February–November 2020), the excess mortality is even higher (83,985 deaths). Noteworthy, the ratio between reported deaths and excess mortality in the period February–November (69%) did not take into account the actual impact of COVID-19 and November’s data are not yet consolidated [44].

### 5.5. Epidemiology in Nursing Homes in Italy

Nursing homes (RSAs) are structures used to take care of elderly people affected by multiple morbidity as well as of not self-sufficient subjects. Other factors that can contribute to worsening the health of guests of these structures, and consequently increase the risk of death, include detachment from affects and the loss of one’s daily life activities, which negatively affect the patient’s mental well-being [45]. After the outbreak of the COVID-19 epidemic in Italy, healthcare residences and post-acute hospitals were probably the most affected care settings, due to both the positivity to the virus of patients and healthcare personnel and the high number of deaths that occurred in a short time. What has happened in recent months and is still happening now has caused enormous uproar and concern in the population and triggered the launch of judicial investigations aimed at identifying any accountability. The high number of victims in nursing homes is probably the result of a rapid contagion in these cohorts of fragile subjects, rather than of actual organizational/structural deficiencies of the structures themselves. [46] A report published by the Lombardy region concerning the analysis of deaths from COVID-19 in the RSAs of the Metropolitan City of Milan highlighted how the mortality within these structures in >70 years of age patients actually increased in 2020 when compared to the previous four years (2016–2019). This increase clearly started in March 2020; peaks of at least 80 deaths per day were registered in the first two weeks of April 2020 compared to an average of about 20 per day in the previous four years. These data are even more significant if we consider that at the beginning of 2020 the average of deaths in this cohort of subjects was lower than the average registered in the 2016–2019 period. The increase in the overall risk of death from 1 January to 30 April 2020 was therefore estimated to be two times higher than the mortality reference value observed in the period 2016–2019. In the period from 1 March to 30 April 2020 the risk of death increased about four times [46]. In the same region, a study analyzed the 2020 mortality data in the provinces of Mantua and Cremona in comparison to the previous two years, evaluating deaths in the general population and in the >75-year-old RSA residents. Although the risk of death within these structures was already higher than that registered in the non-institutionalized population in the pre-COVID-19 era (about 2–3 times in 2018–2019), the study showed that, during the pandemic, the risk of death worsened becoming about seven times greater compared to that of the general population. These estimates were done evaluating and eliminating any bias due to demography and health state [45].

The ISS launched a survey on COVID-19 in RSAs from 24 March 2020 involving 3292 structures (96% of those present on the national territory). At the end of the survey (5 May 2020), approximately 41% (1356) of the structures provided data. The regions that involved more structures were Lombardy (292), Piedmont (249) and Tuscany (200), and in total 97,521 residents were reported as of 1 February 2020 with an average of 72 residents per evaluated facility (range 7–632). [47]

Table 3 reports the total number of RSA residents by region as of 1 February 2020, the total deaths since this date due to any cause, the total deaths due to confirmed COVID-19 and the total number of patients who have died with no diagnosis of COVID-19 but with flu-like symptoms. As of 1 February 2020, 9154 deaths were registered among residents, considering all causes of death; 680 deaths were due to confirmed COVID-19 disease and 3092 due to flu-like symptoms (7.4% and 33.8% of the total deaths occurring within these facilities, respectively) [47].

As noted above, the involvement of nursing homes was linked to not only the contagion of residents but of health workers as well. At the national level, it has been highlighted that, taking into account the survey conducted by the ISS, the regions with the higher rates of structures with infected personnel were the autonomous provinces of Bolzano (50.0%) and Trento (46.7%), followed by Lombardy (40.0%), Piedmont (25.0%), Marche (23.5%), Emilia-Romagna (18.1%), Veneto (16.6%), Liguria (15.8%) Friuli Venezia Giulia (12.8%) and Tuscany (12.4%); values <10% or equal to zero were reported for the other regions. However, this evaluation is affected by the number of swabs performed in each local context. The explanation for this high number of infections and deaths could be related to what has been reported by evaluated structures. Among these, 77.2% reported the lack of Personal Protective Equipment, 33.8% the absence of health personnel, 26.2% the difficulty in isolating residents affected by COVID-19 and 20.9% the lack of adequate instructions on how to cope with the infection. In addition, 12.5% of the evaluated nursing homes had difficulty in transferring residents affected by COVID-19 to hospitals and 9.8% reported a lack of drugs. Finally, 282 nursing homes declared the impossibility of having swabs performed. However, as this last question was included on 8 April 2020, this last datum refers to 52.1% of the structures taking part to the survey (541). Almost all structures decided to block visits to residents by family members and/or caregivers. This measure was implemented by 9 March 2020 by approximately 90% of the structures involved in the survey. Many facilities (68.5%) also reported having resorted to alternative forms instead of visit, such as video calls or phone calls and e-mails [47]. Since 9 March 2020, in the provinces of Mantua and Cremona, in addition to the access ban for visitors, recreational activities were suspended, physical distancing measures were implemented and access to work for staff with symptoms was banned [45]. In addition, the possibility of isolating the patient with suspected or confirmed infection was another measure adopted, even if differently in each structure (single rooms, dedicated structures, etc.). However, 8% of 1351 involved nursing home reported the impossibility of isolating the patient [47].

## 6. Conclusions

The year 2020 will certainly be remembered for the COVID-19 pandemic that endangered our lives, putting a strain on the political, economic and social health balances of the various states of the world and undermining the certainties of millions of individuals. The current epidemiological situation is constantly evolving and remains serious.

An increase in COVID-19 cases was observed in Europe and Italy in the early months of 2021 [48,49]. In Italy, in particular, there was an increase in the incidence of COVID-19 in the younger age groups (0–18 years) and a significant increase in pressure on the prevention departments which led to a delay in notification. and in updating epidemiological data. [48]

Continuing the monitoring of the indicators, also paying more attention to superspreader events by concentrating and intensifying tracking procedures and acting promptly as soon as changes in the epidemiological trend are detected is also fundamental to try to contain the spread of new variants of SARS-CoV-2. The strengths we currently have to keep the epidemiological situation under control are certainly the vaccination campaigns started at the end of last December, the increasing knowledge about SARS-CoV-2, the attention that is paid to the identification of new variants and the implementation of both individual and collective preventive measures. Unfortunately, there are also difficulties: the inability of most European countries to sequence the minimum number of samples required by ECDC, the continuous variation of the spreading dynamics of SARS-CoV-2 and the fatigue of the health systems. The main future issue certainly concerns the possible appearance of new variants with even higher transmissibility and a worse clinical course; this issue is directly linked to the ability of European countries to quickly implement the resources to be dedicated to molecular biology in terms of both equipment and human resources. A second emerging issue is that concerning the duration of immunity in vaccinated individuals. Knowing how long an immunized person is protected from COVID-19 and whether he can still transmit the virus or not is a fundamental step in managing the pandemic from a public health perspective. In addition, as long as the rate of vaccinated individuals does not grow to the target required for the development of herd immunity, it is of fundamental relevance that European countries comply with the indications provided by the ECDC and do their utmost to implement a flawless case surveillance. From this point of view, the greatest difficulties certainly arise when the infections grow rapidly and suddenly, as happened last autumn; therefore, keeping the trend of infections at a stable level or even better to a decreasing level is to be considered a winning point in this phase. Finally, what has emerged during the recent months of the pandemic, in Italy as in the rest of the world, is the awareness of how important it is for a country to have a solid and efficient public health system. This topic, which has been the subject of debate for years, will represent an even more critical point in the near future from which to restart and on which to base health policies at European and global level. The strengthening of health systems as well as the centralization of key issues such as health promotion and prevention must be a priority to avoid a recurrence of an emergency of this magnitude in the coming years.

## Figures and Tables

**Figure 2 ijerph-18-02942-f002:**
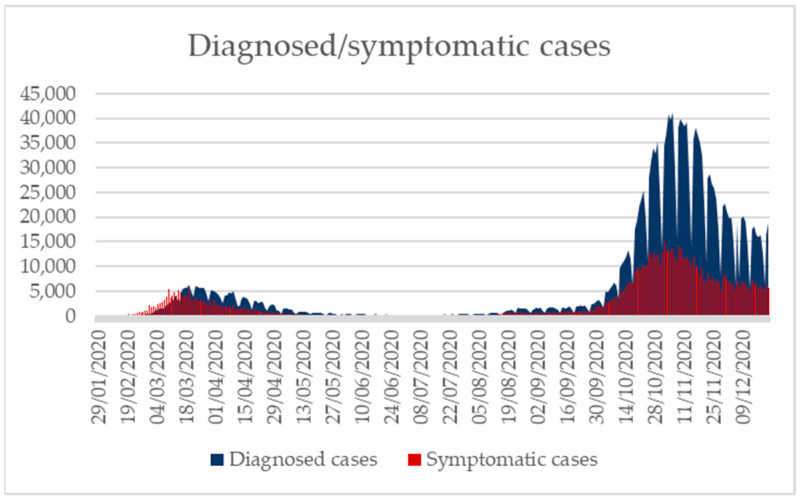
Diagnosed and asymptomatic cases, 29 January 2020–22 December 2020 (modified from [40]).

**Figure 3 ijerph-18-02942-f003:**
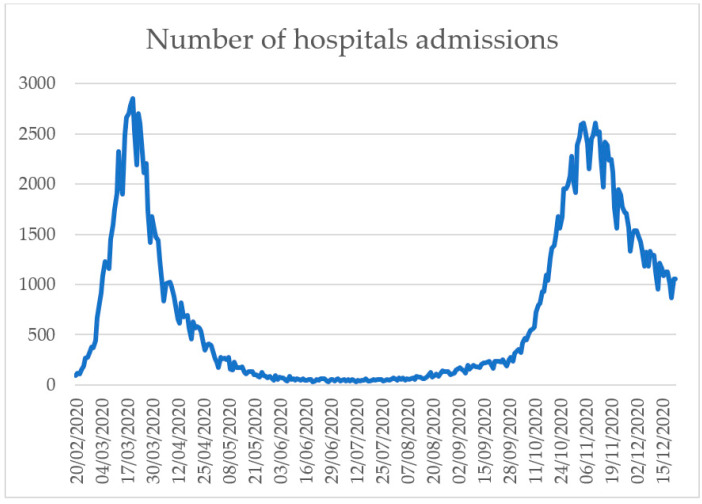
Trend of hospitalizations due to COVID-19, 20 February 20–22 December 2020 (modified from [40]).

**Figure 4 ijerph-18-02942-f004:**
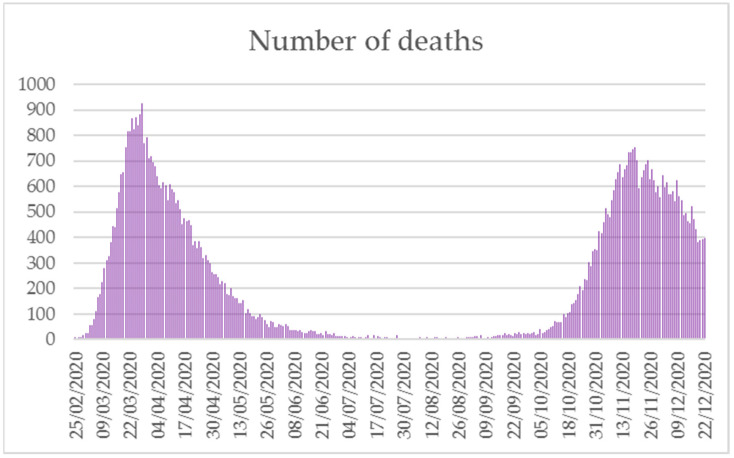
Daily deaths due to SARS-CoV-2, 21 February–22 December 2020 (modified from [40]).

**Table 1 ijerph-18-02942-t001:** ECDC targets of COVID-19 surveillance (modified from [8]).

	Targets	Actions
European Level	Monitor intensity, geographical spreading and disease’s severity	to estimate the impact of the disease on population
	Identify new variants of the virus	to address research for tailored therapies
	Evaluate the impact on health services	to optimize resource allocation
	Evaluate the impact of preventive measures	to evaluate timeline and strength of interventions
National Level	Identify and control outbreaks in the healthcare setting	to protect healthcare workers and patients
	Identify and control outbreaks in families and closed communities	to protect groups at most risk and with the possible worst outcome

**Table 2 ijerph-18-02942-t002:** Regions with higher and lower cumulative incidence per 100,000 inhabitants (modified from [39,41]).

	Region/Autonomous Province	Population Density (Inhabitants/km^2^)	Cumulative Incidence per 100,000
Higher incidence	Aosta Valley	38	5654.14
	AP Bolzano	72	5183.99
	Lombardy	423	4651.55
	Veneto	268	4541.19
	Piedmont	171	4285.39
Lower incidence	Molise	68	2031.00
	Basilicata	55	1771.13
	Sicily	192	1707.49
	Sardinia	68	1529.49
	Calabria	126	1155.61

**Table 3 ijerph-18-02942-t003:** Total of residents in nursing homes involved in the survey on 1 February 2020, total deaths due to all causes, total deaths due to confirmed COVID-19 and total deaths in subjects with flu-like symptoms since 1 February 2020 (modified from [47]).

Region	Total Residents at 1 February 2020	Deaths Due to All Causes (N)	Deaths Due to COVID-19 (N; % *)	Deaths Related to Flu-Like Symptoms (N; % *)
Abruzzo	410	47	1 (2.1)	0 (0)
Bolzano AP	418	28	3 (10.7)	10 (35.7)
Calabria	1510	75	0 (0)	1 (1.3)
Campania	626	50	6 (12)	13 (26)
Emilia-Romagna	7906	639	81 (12.7)	265 (41.5)
FVG	3491	222	6 (2.7)	41 (18.5)
Lazio	4439	158	1 (0.6)	28 (17.7)
Liguria	1515	136	20 (14.7)	34 (25)
Lombardy	26,981	3793	281 (7.4)	1807 (47.6)
Marche	1280	160	13 (8.1)	59 (36.9)
Molise	228	24	0 (0)	2 (8.3)
Piedmont	16,629	1658	161 (9.7)	410 (24.7)
Puglia	2056	111	0 (0)	4 (3.6)
Sardinia	568	67	0 (0)	17 (25.4)
Sicily	930	73	0 (0)	11 (15.1)
Tuscany	9245	640	36 (5.6)	154 (24.1)
Trento AP	1189	99	33 (33.3)	45 (45.5)
Umbria	719	38	0 (0)	11 (28.9)
Veneto	17,381	1136	38 (3.3)	180 (15.8)
TOTAL	97,521	9154	680 (7.4)	3092 (33.8)

AP, Autonomous Province; FVG, Friuli-Venezia-Giulia; ***** the rates reported for each region are calculated on the total of deaths due to all causes in the region itself.

## Data Availability

No new data were created or analyzed in this study. Data sharing is not applicable to this article.

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
