# Peer review of "The Epidemiological Characteristics of the COVID-19 Pandemic in Europe: Focus on Italy"

_ijerph, 2021, doi:10.3390/ijerph18062942_

Round 1
Reviewer 1 Report
Well-written summary of the official epidemiological situation in Europe and in Italy, but currently offers little discussion about it. The manuscript is well structured but the main message this review is offering is not too clear and definitively not novel- it mainly summarizes official surveillance reports which tell a story that is already well known. The manuscript will gain insight by incorporating a critical and constructive analysis about the current surveillance system, differences among countries or regions, i.e. about case definition, indicators (particularly interpretation of results given potential confounders and bias, including denominator data) as well as perspective about the possibility of recommending to countries/intergovernmental organisations to incorporate the epidemiological indicators mentioned at the beginning of the paper to the current parameters to measure the extent and impact of COVID-19 across countries.
Some specific comments:
- L96- the non-compliance rather than the compliance with preventive regulations
- Table 1. Maybe add two columns under European and National Level to explain what does it means in bold and italics (target?) and in plain text (utility?)
- L200- “the infection is not yet defined as zoonoses” I cannot agree with this sentence when there has been evidence of infection both from owners to pets and zoo-keepers to lions as well as from minks to humans. It is even included as an example in the WHO webpage on zoonoses. See for example:
https://science.sciencemag.org/content/371/6525/172
- L205 and L212 that finishes in L213 are missing a reference.
- COVID attributed deaths or all?
- Discuss Italy’s results taking into account at least population density and surveillance efforts. Any other discussion as to why results could differ among regions are welcome too (i.e commercial activity, social habits, compliance with regulations…). Similar when reporting about population characteristics, i.e could women with dementia be confounded by age/lower immunity? Are there any statistics regarding the differences in the descriptive summaries? How many age groups were considered and which?
- Epidemiology in nursing homes in Italy. Lines 363-365 states the number of victims is probably a result of rapid contagion in fragile subjects, rather than because of the structures (is this an opinion of the authors? If so specify, otherwise reference), but in L411-412 one can read that the explanation could be related to what has been reported by evaluated structures. Then, probably it was a combination of both.
- Conclusions to direct further actions include improving case recognition, contact tracing system or recognition of super-spreader events, but none of these have been discussed in the review regarding how it is performed now, identification of gaps or problems and how can these be solved or improved.
Author Response
Sir,
we thank very much for your comments. Please find the following detailed response to each comment.
Well-written summary of the official epidemiological situation in Europe and in Italy, but currently offers little discussion about it. The manuscript is well structured but the main message this review is offering is not too clear and definitively not novel- it mainly summarizes official surveillance reports which tell a story that is already well known. The manuscript will gain insight by incorporating a critical and constructive analysis about the current surveillance system, differences among countries or regions, i.e. about case definition, indicators (particularly interpretation of results given potential confounders and bias, including denominator data) as well as perspective about the possibility of recommending to countries/intergovernmental organisations to incorporate the epidemiological indicators mentioned at the beginning of the paper to the current parameters to measure the extent and impact of COVID-19 across countries.
Thanks for your suggestions. We certainly hope that our manuscript can be a hint for countries/intergovernmental organizations. We have not widely discussed about the current Italian surveillance system because, excluding little differences between Italian regions, the main national indication has been to consider parameter such as R0, Rt (as we described in the second paragraph) and a unique case definition (based on WHO indications). Regarding the differences between European countries, we have not discussed about it because we considered the ECDC’s surveillance system that collects data from European countries systematically and periodically (paragraph 2 and table 1).
Some specific comments:
L96- the non-compliance rather than the compliance with preventive regulations.
Thank you for your suggestion, we modified the sentence.
Table 1. Maybe add two columns under European and National Level to explain what does it mean in bold and italics (target?) and in plain text (utility?)
Thanks for pointing this out. We modified table 1.
L200- “the infection is not yet defined as zoonoses” I cannot agree with this sentence when there has been evidence of infection both from owners to pets and zoo-keepers to lions as well as from minks to humans. It is even included as an example in the WHO webpage on zoonoses. See for example: https://science.sciencemag.org/content/371/6525/172
Thank you for pointing this out. We modified the sentence.
L205 and L212 that finishes in L213 are missing a reference.
Thank you, done.
COVID attributed deaths or all?
Thanks for pointing this out. If it refers to paragraph 5.4 (mortality and lethality), it is COVID attributed deaths.
Discuss Italy’s results taking into account at least population density and surveillance efforts. Any other discussion as to why results could differ among regions are welcome too (i.e commercial activity, social habits, compliance with regulations…). Similar when reporting about population characteristics, i.e could women with dementia be confounded by age/lower immunity? Are there any statistics regarding the differences in the descriptive summaries? How many age groups were considered and which?
Thanks for your suggestions, we added the population density. In our opinion discussion on differences between social habits, commercial activity, etc. of the various region is beyond the scope of our manuscript and for this reason we would like to not modify this part.
Data on preexisting pathologies were taken from a small sample of deceased patients (5,926) after hospitalization. All age groups were included in the analysis.
Epidemiology in nursing homes in Italy. Lines 363-365 states the number of victims is probably a result of rapid contagion in fragile subjects, rather than because of the structures (is this an opinion of the authors? If so specify, otherwise reference), but in L411-412 one can read that the explanation could be related to what has been reported by evaluated structures. Then, probably it was a combination of both.
Thank you, we added the reference.
Conclusions to direct further actions include improving case recognition, contact tracing system or recognition of super-spreader events, but none of these have been discussed in the review regarding how it is performed now, identification of gaps or problems and how can these be solved or improved.
Thanks for pointing this out. Conclusion paragraph has been completely rewritten.
We hope to have taken into account most of the points raised and that our revised manuscript is now suitable for publication.
Thank you
Best regards,
Giovanni Gabutti
Reviewer 2 Report
The paper presents a concise and informative description regarding COVID-19 pandemic in Italy. It is divided into three main parts: description of epidemiological parameters, description of the pandemic in Europe and description of the pandemic in Italy. The text is written mostly clearly, with only several unclear phrases. Personally, I found the manuscript informative and easy to read, especially its first part. Yet, many facts are presented it the text that are not being supported by any reference, also there are several other issues that I have found:
- Line 55-58: Please provide a reference.
- Line 58-59: Please provide a reference.
- Line 72-76: Please clarify whether this paragraph considers a general disease case, COVID-19 pandemic, or the pandemic in Europe/Italy.
- Line 83-86: Please provide a reference.
- Line 86: Please correct COVID19 to COVID-19
- Line 89-90: Please provide a reference.
- Line 122-126: Please provide a reference.
- Line 151-153: The presented sentence is not clear. Please clarify the timeline presented for Italy and other countries, because presented percentage data comes from two different periods of time. Please clarify if “% of cases” refers to “% of cases in Europe”.
- Line 171-173: Please provide a reference.
- Line 183: The word “therefore” should be removed
- Line 183-187: Please provide a reference.
- Line 195-197: The sentence is not clear. Why only 8 of thousands mutations are studied? The sentence implies that these 8 strains also do not present any additional threat; please clarify that.
- Line 204-208: Please provide a reference.
- Line 228: Please paraphrase the sentence ”Fortunately, most cases involve <60-year-old subjects”. I understand the intention behind it, but it sounds to me like it is good that younger people were infected.
- Line 234-236: Please provide a reference.
- Line 236-237: Please provide a reference.
- Line 285: Please correct comma to a decimal point in “5,183,99”
- Line 278-300: I suggest to change the order of the paragraphs, so the early stages would be described first and the late/summary would be as the second paragraph.
- Line 309-310: A biphasic distribution, by definition, has two peaks. Also the presented time lines are not well corresponding (see example). Please paraphrase, e.g.: “The distribution is biphasic with the first peak in the early stages of epidemic (March), while the second is in November.”
- Line 320-329: Please provide a reference.
- Line 330-332: Please provide a reference.
- Line 330-337: I would recommend joining these two paragraphs together.
- Line 385-390: Please provide a reference.
- I would consider joining figure 3 & 4 into a single figure of two separate distributions (e.g. A&B)
Author Response
Sir,
we thank very much for your comments. Please find the following detailed response to each comment.
The paper presents a concise and informative description regarding COVID-19 pandemic in Italy. It is divided into three main parts: description of epidemiological parameters, description of the pandemic in Europe and description of the pandemic in Italy. The text is written mostly clearly, with only several unclear phrases. Personally, I found the manuscript informative and easy to read, especially its first part. Yet, many facts are presented it the text that are not being supported by any reference, also there are several other issues that I have found:
Line 55-58: Please provide a reference; Line 58-59: Please provide a reference.
Thank you, done.
Line 72-76: Please clarify whether this paragraph considers a general disease case, COVID-19 pandemic, or the pandemic in Europe/Italy.
Thank you for your suggestion, we modified the sentence.
Line 83-86: Please provide a reference; Line 86: Please correct COVID19 to COVID-19; Line 89-90: Please provide a reference; Line 122-126: Please provide a reference.
Thank you, done.
Line 151-153: The presented sentence is not clear. Please clarify the timeline presented for Italy and other countries, because presented percentage data comes from two different periods of time. Please clarify if “% of cases” refers to “% of cases in Europe”.
Thanks for pointing this out. We modified the sentence.
Line 171-173: Please provide a reference; Line 183: The word “therefore” should be removed; Line 183-187: Please provide a reference.
Thank you, done.
Line 195-197: The sentence is not clear. Why only 8 of thousands mutations are studied? The sentence implies that these 8 strains also do not present any additional threat; please clarify that.
Thanks for pointing this out. We modified the sentence.
Line 204-208: Please provide a reference.
Thank you, done.
Line 228: Please paraphrase the sentence “Fortunately, most cases involve <60-year-old subjects”. I understand the intention behind it, but it sounds to me like it is good that younger people were infected.
Thanks for pointing this out. We modified the sentence.
Line 234-236: Please provide a reference; Line 236-237: Please provide a reference.
Thank you, done.
Line 285: Please correct comma to a decimal point in “5,183,99”
Thank you, done.
Line 278-300: I suggest to change the order of the paragraphs, so the early stages would be described first and the late/summary would be as the second paragraph.
Thank you for your suggestion, we changed the order.
Line 309-310: A biphasic distribution, by definition, has two peaks. Also the presented time lines are not well corresponding (see example). Please paraphrase, e.g.: “The distribution is biphasic with the first peak in the early stages of epidemic (March), while the second is in November.”
Thank you, we modified the sentence.
Line 320-329: Please provide a reference; Line 330-332: Please provide a reference.
Thank you, done.
Line 330-337: I would recommend joining these two paragraphs together.
Thank you, done.
Line 385-390: Please provide a reference.
Thank you, done.
I would consider joining figure 3 & 4 into a single figure of two separate distributions (e.g. A&B).
Thank you for your suggestion, but we prefer to keep them separate because the deaths did not necessarily occur in hospital nor were secondary to hospitalization.
We hope to have taken into account most of the points raised and that our revised manuscript is now suitable for publication.
Thank you
Best regards,
Giovanni Gabutti
Reviewer 3 Report
(1) The structure of the abstract doesn't follow the standard format, that is with subsections such as: study background, objectives, methods, results, conclusion
(2) The Introduction is too short. There's need to elaborate so as provide the followings: problematic, research question, study interest
(3) Lines 35 - 40: this paragraph must be part of the abstract
(4) It is mandatory to re-organize the manuscript in a comprehensive way:
* Abstract
* Keyswords
* Introduction
* Material and Methods (or Methodology)
* Results
* Discussion
* Conclusion
* References
(5) As it is, it is difficult to know where the data derive from or how they were collected and managed. We assume that from line 41 to line 105, it is about methodology: so, please synthetize and come out with an appropriate and exciting paragraph
Author Response
Sir,
we thank very much for your comments. Please find the following detailed response to each comment.
(1) The structure of the abstract doesn't follow the standard format, that is with subsections such as: study background, objectives, methods, results, conclusion
(2) The Introduction is too short. There's need to elaborate so as provide the followings: problematic, research question, study interest
(3) Lines 35 - 40: this paragraph must be part of the abstract
(4) It is mandatory to re-organize the manuscript in a comprehensive way: * Abstract * Keyswords * Introduction * Material and Methods (or Methodology) * Results * Discussion * Conclusion * References
(5) As it is, it is difficult to know where the data derive from or how they were collected and managed. We assume that from line 41 to line 105, it is about methodology: so, please synthetize and come out with an appropriate and exciting paragraph
We are sorry that our manuscript has not been positively considered and we thank for all the suggestions. However, we would like to point out that this paper is a review. For this reason, we would like to not reorganize the structure of the article, following what has already been done in other already published papers in the same Special Issue.
As suggested, we incorporated lines 35-40 into the abstract.
We hope to have taken into account most of the points raised and that our revised manuscript is now suitable for publication.
Thank you
Best regards,
Giovanni Gabutti
Round 2
Reviewer 1 Report
Thank you for considering my suggestions and going beyond them too. The manuscript has benefited of a deeper insight although I still would have wished to read more about what could be improved in the surveillance systems across Europe and even across Italy although I understand this was out of the scope of this review summary. The guidelines are there at the national or at the European level, but tracking suspects seems to vary widely across areas. Nonetheless and all in all, the improved conclusions mention important elements to consider in the near future that should not be forgotten to be able to fight transboundary infectious diseases globally.
Author Response
10/03/2021
Sir,
we thank very much for your comments.
Please find the following detailed response to each comment.
Thank you for considering my suggestions and going beyond them too. The manuscript has benefited of a deeper insight although I still would have wished to read more about what could be improved in the surveillance systems across Europe and even across Italy although I understand this was out of the scope of this review summary. The guidelines are there at the national or at the European level, but tracking suspects seems to vary widely across areas. Nonetheless and all in all, the improved conclusions mention important elements to consider in the near future that should not be forgotten to be able to fight transboundary infectious diseases globally.
We appreciated your suggestions that significantly improved the quality of our manuscript. Thank you very much for your comments.
Best regards,
Giovanni Gabutti
Reviewer 2 Report
In their respond the Authors say that introduced changes into the manuscript. Nonetheless, several of them, regarding including a reference, are missing in the version I received, as follows from my previous review:
Line 55-58: Please provide a reference; (new line 60-64)
Line 58-59: Please provide a reference. (new line 65-65)
Line 83-86: Please provide a reference; (new line 89-92)
Line 89-90: Please provide a reference; (new line 96-97)
Line 122-126: Please provide a reference. (new line 130-134)
Author Response
10/03/2021
Sir,
we thank very much for your comments.
Please find the following detailed response to each comment.
In their respond the Authors say that introduced changes into the manuscript. Nonetheless, several of them, regarding including a reference, are missing in the version I received, as follows from my previous review:
Line 55-58: Please provide a reference; (new line 60-64)
Line 58-59: Please provide a reference. (new line 65-65)
Line 83-86: Please provide a reference; (new line 89-92)
Line 89-90: Please provide a reference; (new line 96-97)
Line 122-126: Please provide a reference. (new line 130-134)
Thank you for your valuable contribution: due to possible layout errors, the bibliography was not indicated correctly. We have made the required corrections, thank you very much for your advice.
We hope that our revised manuscript is now suitable for publication.
Thank you.
Best regards,
Giovanni Gabutti
Reviewer 3 Report
Dears authors,
If possible, update figures 1 and 2
Author Response
10/03/2021
Sir,
we thank very much for your comments.
Please find the following detailed response to each comment.
Dears authors, If possible, update figures 1 and 2
Thank you for your suggestion. However, updating Figure 1 and 2 would imply to change the purpose of the manuscript that was to give a general framework regarding the epidemiological situation in 2020.
Following your advice we added a short update in the conclusions. Thank you.
We hope that our revised manuscript is now suitable for publication.
Thank you .
Best regards,
Giovanni Gabutti